# A question of scale: modelling biomass, gain and mortality distributions of a tropical forest

Nikolai Knapp[1,3], Sabine Attinger[2,4], Andreas Huth[1,5,6]

[1]Department of Ecological Modelling, Helmholtz Centre for Environmental Research – UFZ, Leipzig, 04318, Germany
[2]Department of Computational Hydrosystems, Helmholtz Centre for Environmental Research – UFZ, Leipzig, 04318, Germany
[3]Thünen Institute of Forest Ecosystems, Eberswalde, 16225, Germany
[4]Institute of Environmental Sciences and Geography, University of Potsdam, Potsdam, 14476, Germany
[5]German Centre for Integrative Biodiversity Research (idiv), Halle-Jena-Leipzig, 04103, Germany
[6]Institute of Environmental Systems Research, University of Osnabrück, Osnabrück, 49076, Germany

*Correspondence to*: Nikolai Knapp (nikolai.knapp@thuenen.de)

**Abstract.** Describing the heterogeneous structure of forests is often challenging. One possibility is to analyse forest biomass in different plots and to derive plot-based frequency distributions. However, these frequency distributions depend on the plot size and thus are scale dependent. This study provides insights about transferring them between scales. Understanding the effects of scale on distributions of biomass is particularly important for comparing information from different sources such as inventories, remote sensing and modelling, all of which can operate at different spatial resolutions. Reliable methods to compare results of vegetation models at grid scale with field data collected at smaller scales are still missing.

The scaling of biomass and variables, which determine the forest biomass, was investigated for a tropical forest in Panama. Based on field inventory data from Barro Colorado Island, spanning 50 ha over 30 years, the distributions of aboveground biomass, biomass gain and mortality were derived at different spatial resolutions, ranging from 10 to 100 m. Methods for fitting parametric distribution functions were compared. Further, it was tested under which assumptions about the distributions a simple stochastic simulation forest model could best reproduce observed biomass distributions at all scales. Also, an analytical forest model for calculating biomass distributions at equilibrium and assuming mortality as a white shot noise process was tested.

Scaling exponents of about -0.47 were found for the standard deviations of the biomass and gain distributions, while mortality showed a different scaling relationship with an exponent of -0.3. Lognormal and gamma distribution functions fitted with the moments matching estimation method allowed for consistent parameter transfers between scales. Both forest models (stochastic simulation and analytical solution) were able to reproduce observed biomass distributions across scales, when combined with the derived scaling relationships.

The study demonstrates a way how to approach the scaling problem in model-data comparisons, by providing a transfer relationship. Further research is needed for a better understanding of the mechanisms that shape the frequency distributions at the different scales.

## 1 Introduction

Forests are complex, heterogeneous ecosystems with characteristics, which can be measured at different scales. They are inherently dynamic systems which are influenced by climate and disturbances (Lewis et al., 2015). In any given forest ecosystem, biomass is variable in space and time, driven by the interplay of productivity and mortality (Rutishauser et al., 2019). Different approaches are being used to quantify forest biomass stocks and changes. These approaches include forest inventory, eddy flux measurements, remote sensing and forest modeling (Mitchard, 2018; Shugart et al., 2015).

The listed methods for quantifying forest biomass often operate at different spatial scales, regarding their extents and resolutions. Inventories measure and map forests at the resolution of individual trees. From the individuals, biomass per area units can be derived. Inventory plots have extents ranging from a few square meters (e.g., national forest inventories) to several hectares (e.g., research megaplots). Eddy flux towers have typical footprint sizes in the range a few hundred meters (Rebmann et al., 2005). Such measurements are an important information source for deriving forest carbon budgets (Brienen et al., 2015;

Hetzer et al., 2020; Hubau et al., 2020).

Much larger extents than the ones of ground-based measurements can be covered by remote sensing. Remote sensing products are typically gridded maps consisting of square-shaped pixels with side lengths, depending on the sensor platform and type of product. Typical pixel sizes are 1 m for canopy height models, 10 to 30 m for tree cover and disturbance maps, 100 m for biomass and 1 km for productivity maps (Asner et al., 2013; Hansen et al., 2013; Running et al., 2004).

Spatial scale is also important for forest models. Models which simulate the biomass dynamics and carbon balance of forests can have different complexities. At one end of the spectrum, there are spatially explicit, individual-based models which include physiological processes for modelling the effects of environmental drivers on the carbon balance (Shugart et al., 2018). On the other end, there are spatially implicit differential equation models describing the carbon balance in terms of and aggregated biomass gains and losses (Fisher et al., 2008). Also, the spatial resolution of the output of forest models can vary. While global

vegetation models produce results at large scales (global extent and resolutions of, e.g., 0.5 degrees; Maréchaux et al., 2021), gap forest models work at local scales (extents of several hectares and resolutions of forest gap sizes, e.g., 20 m, or individual trees; Fischer et al., 2016). It is an open question how to deal with differences in resolutions of ecological models (Fritsch et al., 2020).

The heterogeneity of a forest landscape can be described by frequency distributions of forest attributes, e.g., biomass. Despite

the differences in measurement areas between the methods, it is common practice to report biomass stocks and carbon fluxes in per hectare units. Having units standardized to one reference scale is important to make values comparable and transferable, but bears the risk of taking the unit t ha$^{-1}$ too literally and neglecting the fact that the actual areas might be of very different size. Since the actual areas, which these values represent, do often deviate from one hectare, it is crucial to consider the scale-dependence of the shapes of distributions. It has been shown that forest biomass distributions become increasingly skewed and

long-tailed, with decreasing plot size (Chave et al., 2003). At small scales these distributions cover a wide range of values from

near 0 t ha$^{-1}$ in gaps to 1000 t ha$^{-1}$ in patches with very large trees. However, at the actual 1-ha scale such extreme values are uncommon to observe and the range of typical values is much more reduced.

Commonly, variance decreases with increasing scale. For independent identically distributed (iid) random variables, the aggregation by a factor n (i.e., taking the mean of n values to derive one aggregated value) results in a reduction of variance
by the factor n, and thus a reduction of standard deviation by a factor $\sqrt{n}$, while the expected value (distribution mean) stays the same (Otto and Day, 2011). More generally, also for non-iid and spatial data, the variance reduces with aggregation, as more and more of the fine scale variance occurs within and not above the aggregation level (Smith and Urban, 1988), which is referred to dispersion variance in geostatistics (Marques and Costa, 2014).

In times where field measurements, remote sensing and model simulations are increasingly used in combination, approaches
for harmonizing the different datasets with regard to spatial resolution are required. Hence, with regard to forest biomass distributions and their temporal dynamics the following problems arise: Field measurements provide us with high resolution information, but for limited extents, while remote sensing can cover large extents with limited resolutions. Forest models make use of both kinds of information, either as input or to validate their output against. But it is often unclear how scale affects observed and simulated distribution (Knapp et al., 2018a; Landsberg, 2003; Rödig et al., 2017; Smith and Urban, 1988).
Methods are needed to account for scale effects in model-data comparisons (Rammig et al., 2018).

In this study, we worked towards an approach for transferring frequency distributions of forest biomass between different scales. Such approaches are needed to compare data with simulation output. We tested it for a tropical rainforest for which we also analyzed the distributions of biomass gain and mortality. In addition, we developed two simple forest models to analyze how they can be applied at different spatial scales. The main questions of the study were: 1) How do frequency distributions
of forest biomass vary with spatial scale and which probability density functions describe them best? 2) How can we transfer between these distributions at different scales? 3) How can simple forest models reproduce these distributions at different scales?

## 2. Material and methods

### 2.1 Field data analysis

The 50-ha forest inventory plot (ForestGEO) on Barro Colorado Island (BCI), Panama, served as a study site (Condit, 1998; Condit et al., 2019, 1995; Hubbell et al., 1999). The inventory comprised single tree measurements of all trees with a DBH ≥ 1 cm (ca. 250,000 trees), including the spatial position of each individual (Condit et al., 2012). Aboveground biomass values of each tree were calculated based on a common allometric equation for tropical trees (Chave et al., 2005). Censuses used in this study comprised data from 1985 to 2015 collected in 5-year intervals. Based on unique tree identifiers and AGB values in
successive censuses, AGB gains per tree were calculated. The inventory was aggregated for square shaped plots with 10-, 20-, 50- and 100-m side lengths. Attributes calculated for each of these plots were standing AGB at each census and AGB gain, loss and mortality for each census interval. AGB loss was the sum of AGB of all trees alive in the census at the beginning of

an interval and dead in the following census. AGB mortality was AGB loss divided by the standing AGB at the beginning of the interval. Due to measurement uncertainties some trees showed negative AGB gains. At the individual tree level these negative AGB gains corresponded to 29.6% of the positive AGB gains. When aggregating AGB gain as the sum of single tree AGB gains the negative proportion would decrease to 12.3% (10-m), 7.2% (20-m), 1.5% (50-m) and 0% (100-m). To obtain unbiased, positive AGB gains a correction was done at the single tree level: All negative AGB gains were set to zero and all positive AGB gains were reduced by 29.6% to maintain the overall AGB gain. AGB gains, losses and mortalities were downscaled to annual values through division by 5 years.

For investigating the scaling behavior of AGB beyond the 50-ha plot, an airborne lidar dataset from 2009 covering the whole 15-km² island (Lobo and Dalling, 2014) was used. AGB was predicted at 100-m resolution from mean top-of-canopy height of a 1-m resolution canopy height model using a power law regression model (Knapp et al., 2020) and aggregated at the 200-, 500- and 1000-m scale. Pixels intersecting the coast line were excluded to have pure forest pixels only.

## 2.2 Frequency distributions

The frequency distributions of all four variables (AGB, gains, losses and mortality) were fitted using the R package 'fitdistrplus' (Delignette-Muller and Dutang, 2015; R Development Core Team, 2014) and using two alternative probability density functions: the 1) gamma and 2) lognormal distribution function. Both distribution types can be described with two parameters, respectively. If the parameters are known, the descriptive statistics of the probability distribution (arithmetic mean and standard deviation SD) can be calculated from established equations and vice versa (Table 1). Two different methods were used to fit the probability density functions to the empirical data using: 1) maximum likelihood estimation (MLE) and 2) moments matching estimation (MME). The optimization criterion for MLE is to find distribution parameters which maximize the likelihood for all observations. MME aims at finding the parameters for which the calculated moments match the empirical moments.

**Table 1. Equations for converting between descriptive statistics and parameters of the gamma and lognormal distributions.**

| Distribution | gamma | lognormal |
|---|---|---|
| Descriptive statistics from parameters | $m = \dfrac{\alpha}{\beta}$ $s = \dfrac{\sqrt{\alpha}}{\beta}$ | $m = e^{\mu + \frac{\sigma^2}{2}}$ $s = \sqrt{\left(e^{\sigma^2} - 1\right) \cdot e^{2\mu + \sigma^2}}$ |
| Parameters from descriptive statistics | $\alpha = \dfrac{m^2}{s^2}$ $\beta = \dfrac{m}{s^2}$ | $\mu = \ln\left(\dfrac{m}{\sqrt{1 + \dfrac{s^2}{m^2}}}\right)$ $\sigma = \sqrt{\ln\left(1 + \dfrac{s^2}{m^2}\right)}$ |

**m = mean, s = standard deviation, α = gamma shape, β = gamma rate, μ = lognormal logmean, σ = lognormal logsd**

## 2.3 Scaling equations

In the following, a general equation for scaling of probability distributions is formulated (based on considerations by Dubayah et al., 1997). Let $\theta$ be a parameter of a probability distribution (e.g., SD, shape, rate, etc.). Let $\lambda$ and $\Lambda$ be the side lengths of plots of two different sizes and $\kappa$ be a scaling exponent characterizing how the parameter $\theta$ changes between scales. The values

of $\theta$ (characterizing the frequency distributions; see Table 1 for examples) at scales $\lambda$ and $\Lambda$ shall be called $\theta_\lambda$ and $\theta_\Lambda$, respectively. Then $\theta_\Lambda$ can be calculated from $\theta_\lambda$ in the following way:

$$\theta_\Lambda = \left(\frac{\Lambda^2}{\lambda^2}\right)^\kappa \cdot \theta_\lambda \tag{1}$$

If values for the parameter $\theta$ are known at different scales, $\kappa$ can be derived via log-log-linearization.

$$\frac{\theta_\Lambda}{\theta_\lambda} = \left(\frac{\Lambda^2}{\lambda^2}\right)^\kappa \tag{2}$$

$$\log_{10}\left(\frac{\theta_\Lambda}{\theta_\lambda}\right) = \kappa \cdot \log_{10}\left(\frac{\Lambda^2}{\lambda^2}\right) \tag{3}$$

$$\log_{10}\left(\frac{\theta_\Lambda}{\theta_\lambda}\right) = 2\kappa \cdot \log_{10}\left(\frac{\Lambda}{\lambda}\right) \tag{4}$$

Hence, in log-log-space, the ratio of $\theta$ at two different scales is a linear function of the ratio of areas with $\kappa$ as slope. If the ratio of side length instead of areas is used, the slope of the relationship becomes $2\kappa$. In the further analyses, the area ratio was used (Eq. (3)). The focus of this study was to investigate how the scaling equation can be used to calculate the SD of forest attribute

distributions at a certain target scale from a known SD at a reference scale, which will be called rescaling further on. A graphical example is given in Fig. 1 for the SD of the biomass distribution using 10 m as the reference scale. The analyses focused on

the scales between 10 and 100 m, where the frequency distributions of forest biomass were considered to be primarily driven by the demographic processes growth and mortality (Smith and Urban, 1988), while at larger scales environmental gradients in heterogenous landscapes lead to deviations from the scaling relationship (lidar analysis).

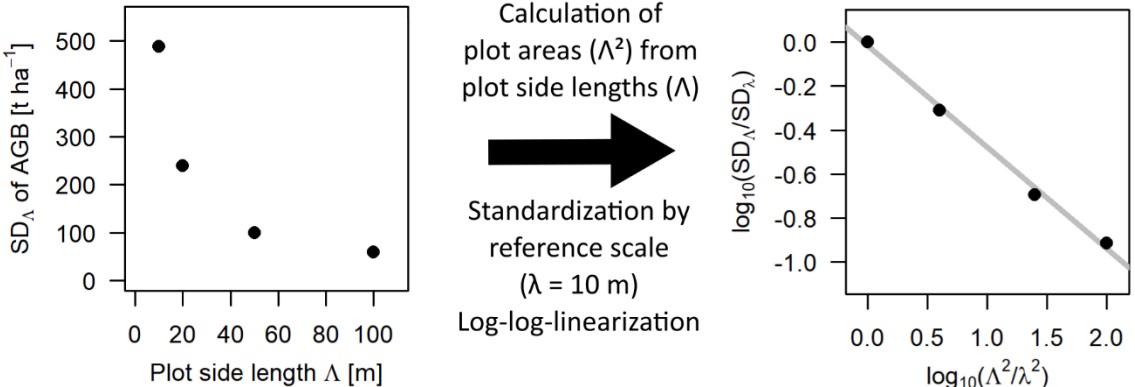

Figure 1. Scale dependence of the standard deviation (SD) of the distribution of aboveground biomass (AGB). The left graphic shows the SD values at each scale ($\Lambda$) plotted over the respective plot side lengths $\Lambda$. The right graphic shows the scaling relationship using plot areas ($\Lambda^2$ instead of $\Lambda$) and after standardizing with 10 m as the reference scale ($\lambda$), and plotting the values in a log-log-linear way. The slope of the grey regression line is the scaling exponent $\kappa$.

## 2.4 Comparing frequency distributions

Two metrics were used to quantify the agreement between different probability density functions (PDFs). The first metric was the relative overlap between PDFs (OVL; Inman and Bradley, 1989), i.e., the intersection of the areas under the curves (AUC) divided by the union of the areas under the curves of two PDFs. The R package 'overlapping' was used for computation of OVL (Pastore and Calcagnì, 2019).

$$OVL = \frac{AUC(PDF_1) \cap AUC(PDF_2)}{AUC(PDF_1) \cup AUC(PDF_2)} \tag{5}$$

The second metric was the relative error of standard deviations (RESD), which we defined as the absolute difference between the SD of an estimated (e.g., rescaled or simulated) $PDF_1$ and an observed (empirical) $PDF_2$, normalized by the SD of the latter.

$$RESD = \frac{|SD(PDF_1) - SD(PDF_2)|}{SD(PDF_2)} \tag{6}$$

Hence, two PDFs are in close agreement, if OVL is close to one and RESD is close to zero. OVL is a good quantitative measure for visual agreement of the main bodies of the distributions. RESD, in contrast, is sensitive to the influence of the extreme values in the tails of the distributions, which is often not apparent from visual inspection of the PDFs.

## 2.5 Forest simulation model

A simple grid-based forest growth model was used to simulate forest dynamics over time and obtain the resulting AGB frequency distribution at mature stage. The model is based on the model suggested by Fisher et al. (2008), in which the change of AGB (B) is described as a differential equation involving an AGB gain parameter (G) and an AGB mortality parameter (M).

$$\frac{dB(t)}{dt} = G - M \cdot B(t) \tag{7}$$

The analytical solution of this equation provides the average trajectory of forest biomass (over a large area) for given G and M and the average mature AGB is G/M.

$$B(t) = \frac{G}{M}(1 - e^{-Mt}) \tag{8}$$

For simulating small scale heterogeneity including patch dynamics, Fisher et al. (2008) used a grid-based approach. Each grid cell represented 10 m × 10 m of forest area. At each simulation time step, they applied Eq. (7) to calculate the change of AGB in each cell, based on a constant AGB gain (G) and a stochastic mortality (M). A mortality event in their case meant that B was set back to zero in a patch and growth restarted from bare ground. Setting a patch back to zero is a valid assumption as long as patches are at the scale of single trees (e.g., 10 m), but using this approach at larger patch scales would rather correspond to large stand replacing disturbances and no longer to natural tree mortality. Hence, in this study, the model was modified such that G and M both follow continuous probability distributions.

The parameters G and M were assumed to be distributed like the AGB gain and AGB mortality values derived from the field inventory. Variability in gain G represents differences in site conditions or growth rates of trees and was assumed to vary in space but stay constant in time. Hence, in the simulation, each patch was assigned a random G drawn from the AGB gain distribution and this G was kept constant for the patch throughout the simulation run. Variability in mortality rate M reflects the stochastic nature of mortality. In a common year, only small amounts of AGB are lost per patch, due to small trees dying from competition. Large AGB losses due to dying large trees are rare events. Hence, mortality rate M was drawn randomly from the AGB mortality distribution for each patch at each time step. This simulation approach was applied at different grid cell sizes (10-, 20-, 50-, 100-m). The total simulation area was set to 25 km² (5 km × 5 km) to obtain smooth output distributions. The simulation time step was one year and 300 years were simulated to reach a mature forest.

The AGB distributions in equilibrium state after 300 years of simulation time were analyzed by comparing them to the AGB distributions from the field inventory. The simulations were conducted with G and M coming from different probability distributions. The probability density functions could either be gamma or lognormal, fitted with either MLE or MME and be derived from the field at either 10-, 20-, 50- or 100-m scale (reference scale). The scaling equation was used to rescale SD and probability density function parameters from one reference scale to the other three scales, to conduct simulations at all four scales for each of the possible input constellations (pre-model scaling). The resulting biomass distributions at the end of the

simulations were aggregated from the finer to the coarser scales (post-model scaling). For evaluating the goodness-of-fit for
the AGB distributions, OVL and RESD were calculated pairwise between each simulated (and aggregated) distribution and
the empirical distribution at the respective scale. For evaluating the simulation run as a whole, the mean OVL and mean RESD
over all single distributions was calculated.

## 2.6 Derivation of biomass distributions from theory – white shot noise

As a second approach, the AGB distribution can also be derived analytically as a function of G and M. This was done to test
if such a direct calculation of the biomass distribution is possible and whether this can be applied at multiple scales as well.
The approach assumes M to be white shot noise. This type of noise is used in environmental modeling to describe pulsed
processes that occur at irregular time intervals (e.g., rainfall at daily resolution). White shot noise assumes pulse intensities
and interval lengths both to be exponentially distributed and is, therefore, characterized by two parameters: the mean intensity
and the mean occurrence probability.

It was assumed that the AGB mortality at small spatial resolution (10 m × 10 m) can be described by white shot noise. Since
observed AGB mortality is never exactly zero, due to small trees dying every year, the mean occurrence probability can be
assumed as one. The mean intensity then directly corresponds to the mean AGB mortality $\bar{M}$. It can be shown that under this
white shot noise assumption the AGB for a given G and $\bar{M}$ value follows a gamma distribution of the following form (Ridolfi
et al., 2011):

$$B \sim Gamma(\frac{1}{\bar{M}} + 1, \frac{1}{G}) \tag{9}$$

For a range of k different G values, which represent the spatial heterogeneity in growth rates and which themselves follow a
gamma or lognormal distribution, AGB of a forest landscape can be described by a mixture of gamma distributions.

$$B \sim \frac{1}{k} \cdot \sum_{i=1}^{k} Gamma(\frac{1}{\bar{M}} + 1, \frac{1}{G_i}) \tag{10}$$

The exponential distribution of mortality intensity, as assumed by the theory, has a monotonically decreasing probability
density function. Such probability distributions are usually only observed for mortality at small scales, while mortality follows
unimodal distributions at larger scales (see e.g., Fig. 4). Hence, for deriving AGB distributions from white shot noise theory
at larger scales, there are different options, of which two were investigated here. 1) The simplest one is to apply the white shot
noise directly at all scales, ignoring the actual shape of the mortality distribution. 2) Alternatively, white shot noise was applied
to derive the AGB distribution only at 10-m scale, and the SD of this distribution was rescaled in a post-model scaling using
Eq. (1) and an empirically derived scaling exponent κ. The rescaled SD was used to approximate the AGB distribution with a
lognormal distribution. The main steps in the analysis, from inventory data to frequency distributions to scaling relationships
and forest model applications at different spatial scales are summarized in Fig. 2.

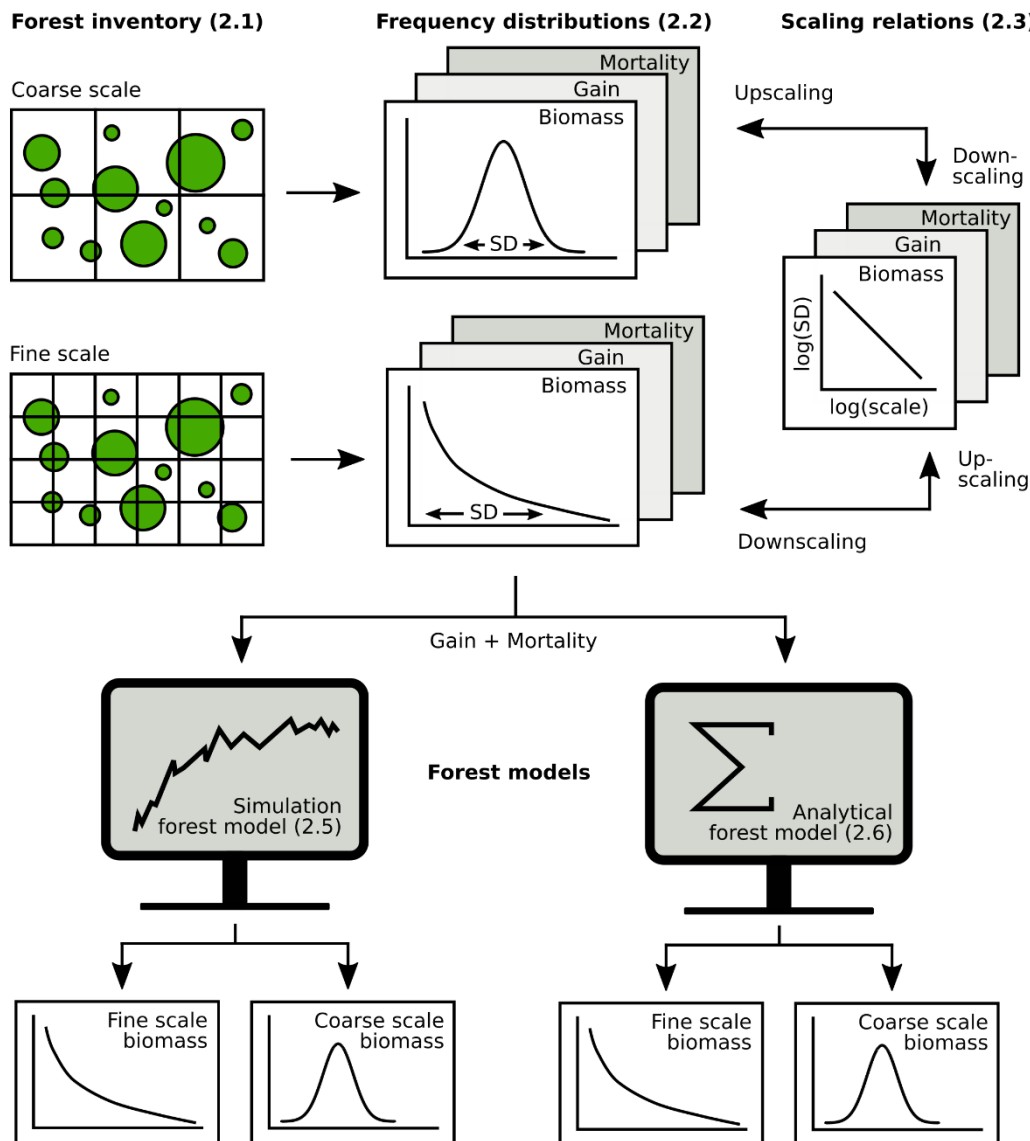

**Figure 2. Overview of the analysis (section numbers in brackets). Forest inventory data was aggregated at different spatial scales to obtain frequency distributions of biomass, gain and mortality. Next, scaling relationships were established for up- and downscaling of frequency distributions. Then, it was tested how biomass distributions at different scales can be derived from gain and mortality using either simulations or an analytical forest model.**

## 3 Results

### 3.1 Scaling relationships of standard deviations of forest attribute distributions

The relationships between spatial scale (plot area) and standard deviations of the frequency distributions were derived for AGB, gain, loss and mortality within the 50-ha forest plot for the scale range from 10 to 100 m. The scaling exponents for AGB ($\kappa$ = -0.461), AGB gain ($\kappa$ = -0.468) and AGB loss ($\kappa$ = -0.467) were similar, whereas it was different for AGB mortality ($\kappa$ = -0.301). The log-log-linear relationships are shown in Fig. 3.

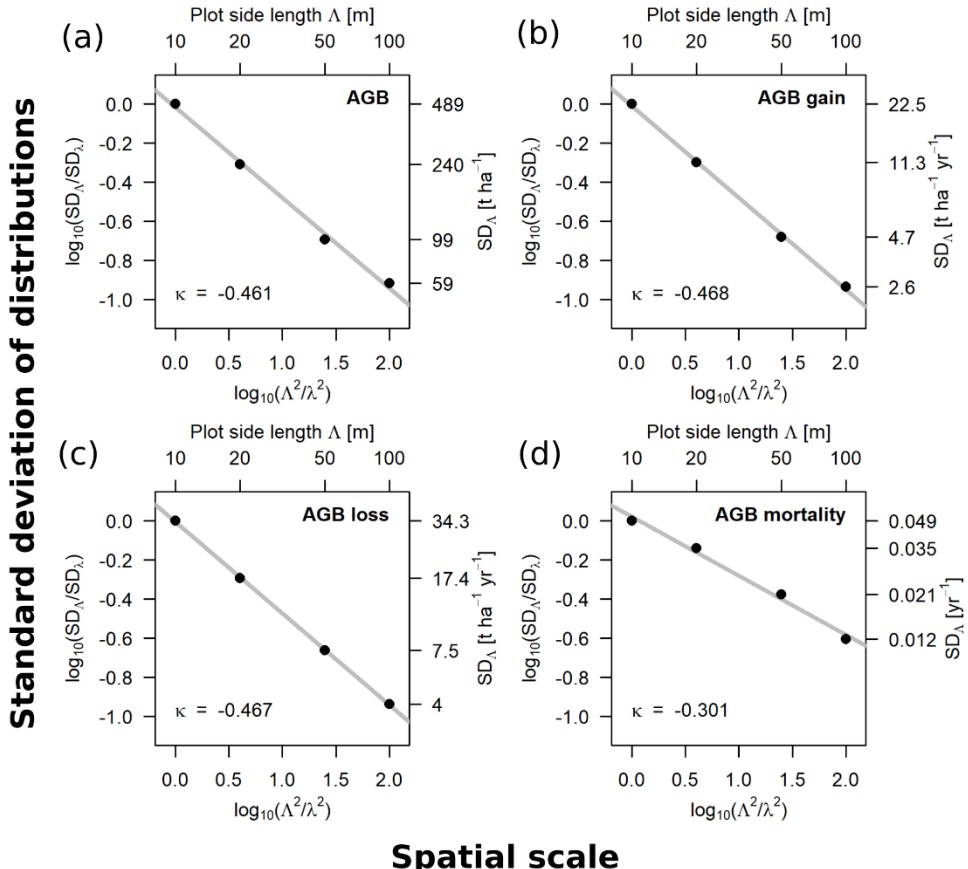

**Figure 3. Scaling relationships for aboveground biomass (AGB) (a), gain (b), loss (c) and mortality (d), derived from the standard deviations (SD) of the empirical distributions. Fitted slopes represent the scaling exponents ($\kappa$ in Eq. (1-3)). Plotted are the log10-transformed ratios of the SDs over the log10-transformed ratios of plot areas, with 10 m being the reference scale ($\lambda$). The top axis shows the plot side lengths ($\Lambda$) and the right axis shows the values of the SD for each scale, respectively.**

Beyond the 50-ha plot, AGB was analyzed via a lidar-derived AGB map (Fig. S4). The lidar analysis demonstrates that at

235 100-m scale the inventory-based SD of the AGB distribution within the 50-ha plot (59 t ha$^{-1}$) matches the lidar-based SD of the whole island (59 t ha$^{-1}$), while the lidar-based SD within the 50-ha plot was somewhat smaller (49 t ha$^{-1}$). At the 200-m

scale, the lidar-based SD within the 50-ha plot (29 t ha$^{-1}$) conforms the scaling relationship. At the 500-m scale, it is somewhat higher than expected (20 t ha$^{-1}$), but at this large pixel size a strict spatial overlap with the 50-ha plot is not possible and the area contributing to this value is 150 ha (6 pixels). The island-wide SDs at 200-m (47 t ha$^{-1}$) and 500-m (37 t ha$^{-1}$) resolution are higher than predicted by the scaling relationship. This additional variability can be explained by the larger extent and possible autocorrelation in the spatial distribution of biomass across the island. At 1000-m scale, the island-wide SD (10 t ha$^{-1}$) is close to the scaling relationship, which can be explained by the fact that only five square kilometers in the center of the island, around the 50-ha plot, were analyzed while all square kilometers intersecting the shore line were excluded.

### 3.2 Fitted distribution functions

For each of the four variables of interest (AGB, gain, loss and mortality) the best describing distribution function was identified, by fitting either lognormal or gamma distribution functions with the maximum likelihood or moments matching method. Selected fits for all four spatial scales (10, 20, 50 and 100 m) are shown in Fig. 4.

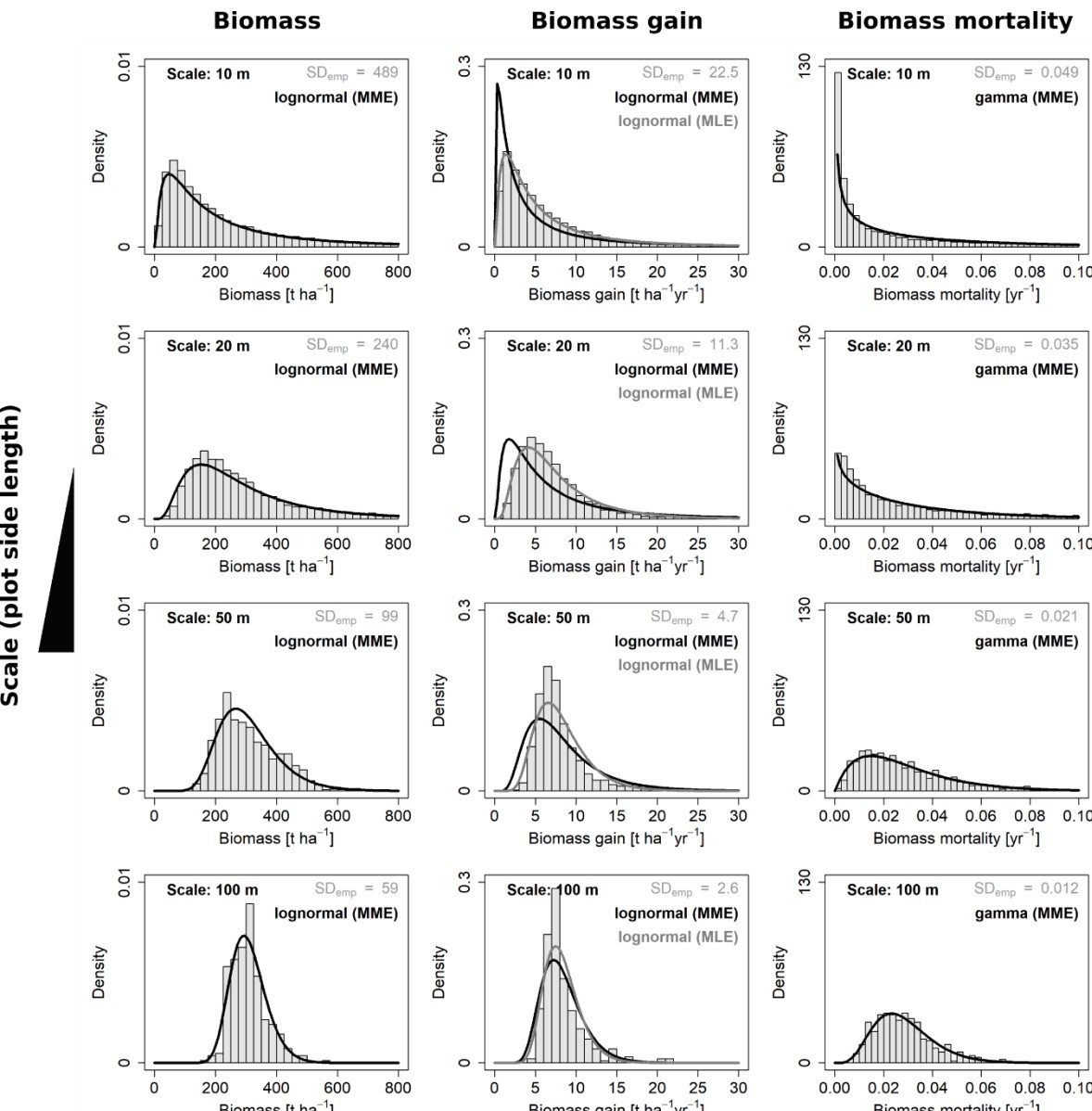

**Figure 4. Frequency distributions for aboveground biomass and its gain and mortality analyzed at four different scales. Shown are the best fitting probability density functions for each of the three variables at each scale and the respective empirical distributions from the field (grey histograms). For AGB gain, each of the two fitting methods (MME and MLE) were superior regarding one criterion, hence, curves are plotted for both.**

From each fit at one scale (reference scale) the expected SDs at all other scales (target scales) were calculated using the scaling equation (Eq. (1)) with the empirically derived scaling exponents κ, respectively. From these rescaled SDs the function parameters and probability density curves were derived using the equations in Table 1. An example for downscaling AGB from 100-m to the three smaller scales is shown in Fig. 5.

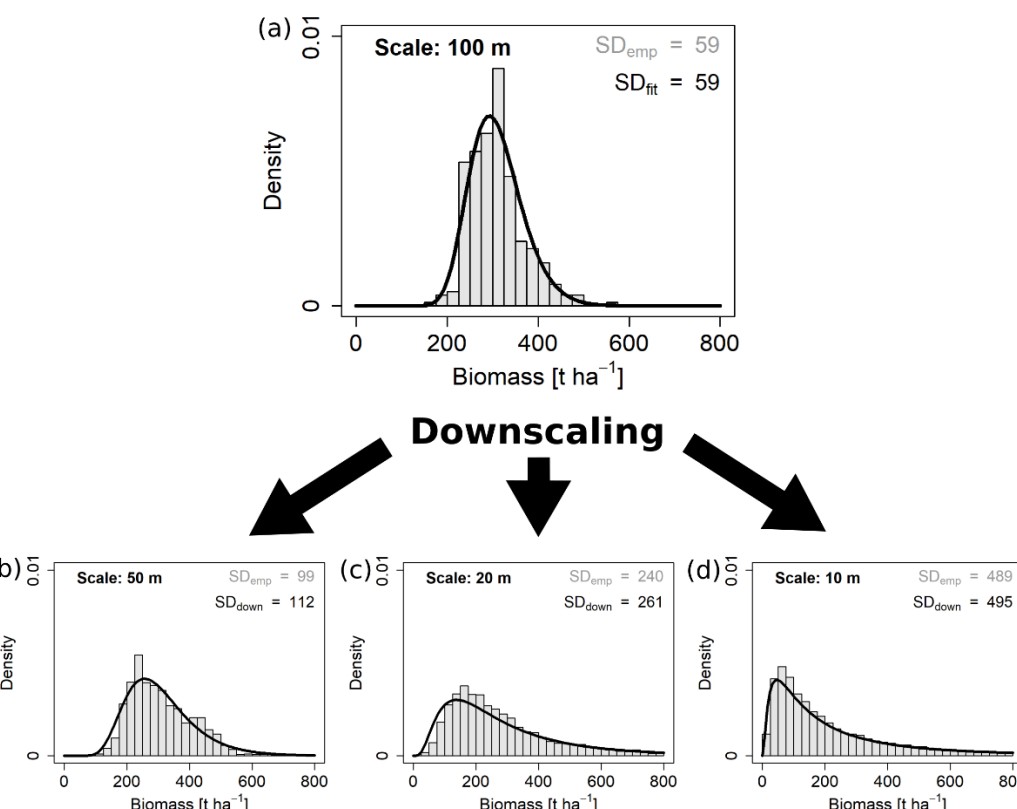

**Figure 5. Downscaling the aboveground biomass distribution from 100-m scale to three other scales. Black curves represent the downscaled (down) lognormal probability density functions and grey histograms the empirical (emp) distributions from the field. Standard deviations are given for each distribution.**

To identify the best fitting distribution and method for each variable, i.e., the most consistent across scales, two criteria were used: 1) the mean overlaps (OVL) between empirical and fitted probability densities and 2) the mean relative error of the SD (RESD). Mean in both cases refers to mean over all four spatial scales. Table 2 lists the values for these criteria for all distributions and methods and Fig. S1 shows the best matching distribution function for each variable at all spatial scales.

**Table 2. Goodness-of-fit criteria values for the different distribution functions and fitting methods for all four variables. Best matching distributions, i.e., those with the largest mean overlap (OVL) and smallest mean relative error of standard deviation (RESD), are highlighted in bold font. Mean hereby refers to the mean over all four spatial scales (10, 20, 50, 100 m).**

| Variable | Distribution | Method | Mean OVL | Mean RESD |
|---|---|---|---|---|
| AGB | lognormal | MLE | 0.883 | 0.08 |
| **AGB** | **lognormal** | **MME** | **0.887** | **0.056** |
| AGB | gamma | MLE | 0.752 | 0.167 |
| AGB | gamma | MME | 0.771 | 0.056 |
| **AGB gain** | **lognormal** | **MLE** | **0.781** | **0.356** |
| **AGB gain** | **lognormal** | **MME** | **0.665** | **0.03** |
| AGB gain | gamma | MLE | 0.636 | 0.366 |
| AGB gain | gamma | MME | 0.567 | 0.03 |
| AGB loss | lognormal | MLE | 0.839 | 0.172 |
| **AGB loss** | **lognormal** | **MME** | **0.877** | **0.013** |
| AGB loss | gamma | MLE | 0.653 | 0.327 |
| AGB loss | gamma | MME | 0.631 | 0.013 |
| AGB mortality | lognormal | MLE | 0.732 | 0.579 |
| AGB mortality | lognormal | MME | 0.699 | 0.052 |
| **AGB mortality** | **gamma** | **MLE** | **0.824** | **0.054** |
| **AGB mortality** | **gamma** | **MME** | **0.82** | **0.052** |

**AGB = aboveground biomass, MLE = maximum likelihood estimation, MME = moments matching estimation**

For three of the four variables, there was a clear best fit according to the criteria (Table 2): AGB (lognormal MME), AGB loss (lognormal, MME) and AGB mortality (gamma, MLE or MME equally good). For AGB gain, however, the best fitting lognormal distribution functions derived from the MLE and MME methods were considerably different from each other (Fig. 4). While the MLE fits had the higher overlap with the field data, they produced large errors when used for calculating the SD at the other scales (large mean RESD). Consistent rescaling of the SD of AGB gain was only achieved when using the MME

fit, albeit the weaker overlap of these distributions with the field data. Comparing Fig. S1b with Fig. S2a illustrates the differences of the methods for AGB gain in detail and Fig S2b+c illustrates the RESD for both.

### 3.3 Simulation results

Simulations with the stochastic forest model were conducted at the scales of 10, 20, 50 and 100 m. The resulting biomass distributions at the end of the simulations were aggregated to the coarser scales, respectively. By testing a range of 64 different

combinations of input distribution functions (G and M) and reference scales (at which these distributions were derived), well performing combinations were identified based on the goodness-of-fit criteria OVL and RESD (Table S1, Fig. S5 to S8). It was found that simulated AGB distributions matched the field distributions particularly well when G and M distributions were derived at 50-m scale (reference scale). Fig. 6 shows the results for the unscaled case at 50-m scale and the rescaled

distributions at the other scales. The distribution functions identified as best for gain and mortality, based on the field data

analyses (Table 2), were also among the best for G and M, based on the simulation analyses (Table S1). If G was modelled as

a lognormal (MLE fit at 50-m scale) and M as gamma (MME fit at 50-m scale) distribution, rescaling and subsequent stochastic

simulations resulted in consistent AGB distributions at all scales (Fig. 6).

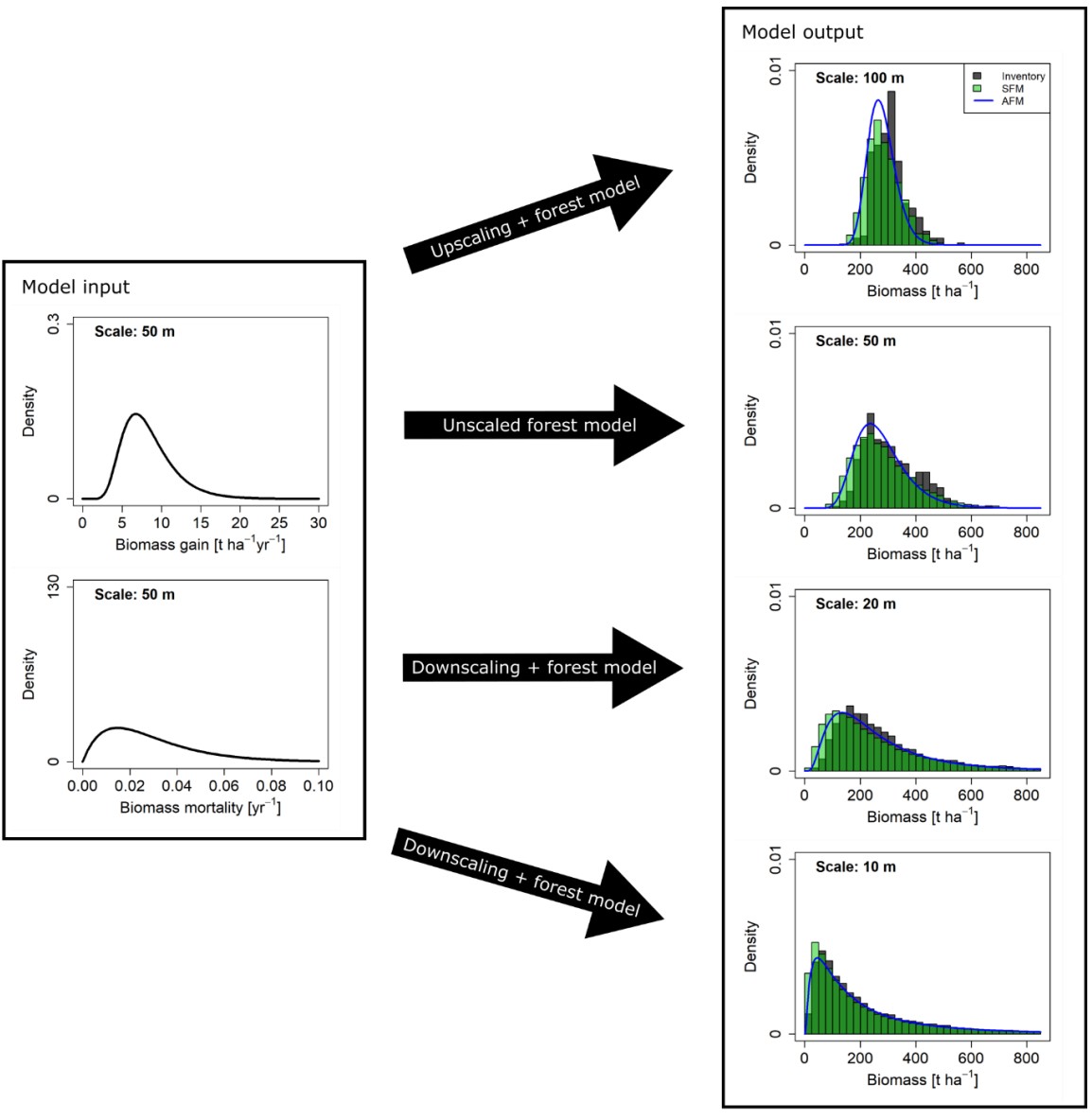

**Figure 6. Biomass distributions at different spatial scales resulting from different approaches. On the left, curves show the gain (G)**
**and mortality (M) distributions used as input. On the right, dark grey histograms show the empirical distributions from the field**
**inventory. Green histograms show the simulation results from the stochastic forest model (SFM). All SFM simulations were driven**
**by the two probability density functions for AGB gain and mortality derived at 50-m scale (left), which were rescaled for simulations**
**at 10-, 20- and 100-m scale. Blue lines show the AGB distribution functions calculated with the analytical forest model (AFM).**

It was found that several versions of the stochastic forest model simulations could produce realistic AGB distributions across
scales (Fig. S5 to S8 provide details about the different reference, simulation and aggregation scales). The best matching AGB
distributions were observed for simulations using 50-m as reference scale (mean OVL = 77%, mean RESD = 9%, Table S2).
At the 20-m reference scale, the largest observed mean OVL was 75% and the smallest observed mean RESD was 12% (albeit,
the two were from different simulations, Table S1). At the 10-m reference scale, the best mean OVL was the smallest among
all reference scales (65%), while the best mean RESD was good (13%, Table S1). At the 100-m reference scale, the best mean
OVL was better (70%), but the best mean RESD was far higher than for any other reference scale (34%, Table S2).

One consistent finding across scales was that simulations using lognormal distributions and the MLE fitting method were more
abundant among the best simulations than the ones using gamma distributions and the MME method. In fact, at any reference
scale, the approach which used only lognormal and MLE for modeling G and M ranked among the best simulations.
Approaches using gamma or MME for more than one distribution (G and M) never ranked among the best (Table S1 and S2).
Examples for simulated AGB distributions at all scales (with OVL and RESD values), when starting from different reference
scales, are shown in Fig. S5 to S8.

## 3.4 White shot noise calculations

As an alternative to the simulations with the stochastic forest model, the calculation of AGB distributions from an analytical
forest model was tested. The mortality was assumed as white shot noise, i.e., only its total mean ($\overline{M} = 0.0285$) was required.
Since the assumption of white shot noise mortality was considered most appropriate at the finest scale, the probability density
function (gamma mixture distribution) was derived at 10-m scale and upscaled with the scaling relationship, using a lognormal
distribution as approximation for the mixture distribution. The distributions resulting from the analytical model (blue lines in
Fig. 6) are close to the field data and the simulation model outputs. A quantitative analysis based on OVL and RESD is
presented in Fig. S9, which also shows how a direct application of white shot noise at larger scales leads to deviations in the
AGB distributions.

## 4 Discussion

It was shown how the distributions of forest biomass, gain and mortality vary with spatial scale. While distributions at small
scale show large SDs due to the large heterogeneity between patches, SDs decrease due to spatial averaging and the loss of
dispersion variance at larger scales. Methods for fitting the distributions best across all scales and transferring their standard
deviations between scales were identified. It was shown how scaling was necessary to reproduce the biomass distributions
with two simple forest models.

## 4.1 Scaling of forest attributes

For transferring between the distributions at the different spatial scales, empirical scaling coefficients were derived. The exponents for scaling the SDs of the distributions were very similar for the extensive properties aboveground biomass stocks, gains and losses (all between -0.46 and -0.47) while the one for the intensive property mortality was considerably different (-0.3). Theory states that the SD of an independent identically distributed random variable changes with the square root of the area ratio between scales, i.e., a scaling exponent of -0.5. The slight deviation from -0.5 is an indication of spatial structure in the data, i.e., that grid cells are not fully independent.

The different scaling exponent of mortality is likely caused by the fact that mortality is a ratio of AGB loss and stock. This characteristic of being a ratio between two extensive properties makes mortality a non-additive intensive property. When aggregating mortality values from smaller to larger scales, the weighted mean has to be used, with AGB as a weighting variable. It further has the consequence that the mean mortality over the whole 50-ha plot is not the same when calculated at different resolutions. The mean mortalities of all 10- and 20-m patches were 0.035 $yr^{-1}$ and 0.031 $yr^{-1}$, respectively and only stabilized above the 50-m scale at 0.029 $yr^{-1}$ (Fig. S1d). All the other attributes in this analysis were additive and had a stable mean across scales.

The lidar-based scaling of AGB beyond the 50-ha plot and across the whole island showed that the SD of the distribution decreases further, but does no longer strictly follow the scaling relationship from within the plot. With increasing scale, the spatial pattern of AGB distribution is possibly increasingly dominated by environmental factors, which in the case of Barro Colorado Island are topographic slope, soil properties and forest age due to disturbance history (Mascaro et al., 2011). These covariates may alter the slopes of scaling relationships. Additionally, the spatial coverages of the AGB maps were different, due to exclusion of pixels intersecting the coast line, which led to a small sample size of only five pixels at the largest analyzed scale (1000-m). Further analyses could investigate the scaling in heterogenous landscapes, e.g., based on synthetically generated landscapes with spatial autocorrelation or based on regional or global remote sensing products. With such approaches, an invariability-area-relationship has been suggested (Wang et al., 2017).

## 4.2 The choice of the proper distribution fit

Parametric probability density functions represent idealized models to which real empirical data may either conform well or not. To fit the parameters of distribution functions to match the data, different target criteria can be formulated, such as maximizing the likelihood of all observed values (maximum likelihood estimation) or matching the moments of the empirical data (moments matching estimation). Thus, the choice of the probability density function (lognormal or gamma) and fitting method (MLE or MME) affects the estimated parameters and hence SD. It was found that for the majority of forest attributes (AGB, loss, mortality) a best fitting distribution function, which was also consistent across scales, could be identified. The example of AGB gain, however, demonstrated that different methods for distribution fitting can have different outcomes and that neither lognormal nor gamma distribution functions matched the empirical distribution well across scales. Maximum

likelihood estimation usually results in high overlaps of empirical and fitted distributions. However, it does not necessarily lead to a good match of the moments of empirical and fitted distribution. Because of this, SDs of the fitted probability density functions deviated from the empirical SDs, and they deviated differently at each scale. This led to inconsistent probability density functions when applying the scaling relationship for rescaling of the SDs. With moments matching estimation, SDs were correctly estimated at all scales and could consistently be rescaled. The price for the matching moments were lower overlaps between empirical and fitted distributions.

### 4.3 Scaling of forest models

It was found that both forest models could reproduce the biomass distributions observed in the field at different spatial scales. For the simulation model, it proved to be best, to choose an intermediate reference scale (50-m) for deriving the information about gain and mortality and using a pre-model scaling to drive simulations at the different other scales. In this way, the simulation model could reproduce the biomass distributions at the four simulation resolutions.

Choosing the lower (10-m) or upper (100-m) end of the scale range as reference scale led to less agreement of the simulated and the field distributions. An explanation for this might be that more bias is introduced when rescaling over a longer scale range, rather than from an intermediate scale which is closer to both ends. Additionally, using the small scales as reference scale is probably problematic, because mean mortality is positively biased at these scales, due to the non-additivity of mortality discussed earlier. Simulations with 100-m reference scale resulted in too large SDs of the distributions across all scales, which can be explained by the fact that the AGB gain distribution underlying the simulations was most difficult to fit and rescale from the 100-m reference scale (Fig. S2a). The reason why this was the case remains speculative, but could be explained by the effect that the heterogeneity caused by the forest structure itself (tree positions and sizes) becomes comparatively small at 100-m scale, such that underlying landscape gradients (topography, soil conditions) make the distributions wider than expected from forest dynamics alone (Mascaro et al., 2011).

The simulation model produced the most consistent results when the input distributions were fitted using maximum likelihood estimation. This was unexpected, considering the earlier finding that moments matching estimation allows for more consistent scaling of SDs based on the field data analysis. An explanation could be that, while moments matching fits are more accurate with regard to the SD, maximum likelihood fits are overall "closer" to the real distributions (higher distribution overlap). The deviation of moments matching fits from the real distributions (smaller distribution overlap) may pose a problem in simulations where these distributions are used for drawing random numbers many times. Maximum likelihood fits might be more appropriate in this case, especially if they were obtained at intermediate scales, from where distribution parameters, like SD, can be rescaled in both directions without causing much bias.

A fast solution for approximating the AGB distribution of the forest from the AGB gain and mortality distributions was given with the analytical model using white shot noise. This approach does not require iterative simulation over time. However, it requires the assumption that mortality follows an exponential distribution, which is a simplification. At the 10-m scale the mortality distribution is at least more similar to an exponential distribution than at the larger scales. Indeed, assuming white

shot noise directly at large scales did not approximate the AGB distributions well. But, assuming white shot noise only at the 10-m scale and applying a post-model upscaling, enabled a good approximation of the AGB distribution across all scales without running a simulation. This is an example of an inherent scale of a model, i.e., the method white shot noise cannot be applied at arbitrary scales, but scaling helps to apply the process at its inherent scale and nevertheless produce results at other scales.

Thus, it was found that the analytical model worked best when applied at 10-m scale, where the assumption of a white shot noise mortality was the most justified, while the simulation model worked best with parameters derived at 50-m scale, from where the least bias was introduced when rescaling the parameters. Such optimal scales always depend on the considered processes and need to be identified in every model.

## 4.4 Perspectives

Quantifying the carbon budget of forests is one of the most important reasons for combining forest models with remote sensing (Shugart et al., 2015). Increasingly, the two techniques are being combined for this purpose, either using remote sensing information for model parameterization, initialization, calibration or validation or using model information for remote sensing interpretation (Plummer, 2000). In this context, it is important that remote sensing products and models describe variables at the same spatial resolution. In case they are operating at different resolutions, appropriate scaling of the variables needs to be conducted.

Descriptive statistics of distributions and relationships between variables may critically depend on the chosen aggregation approach, a phenomenon long known as the modifiable areal unit problem (MAUP; Openshaw, 1983). The MAUP has been divided into the zoning and scaling problem. While the former refers to the problem of where to draw the borders between aggregation units, the latter refers to the size of the aggregation units, which is particularly relevant for regularly gridded data and grid-based simulation models (Wong, 2008). It can only be avoided by identifying basic ecological meaningful entities (trees in forests) and conducting analyses and model simulations exclusively at their level (Jelinski and Wu, 1996). This, however, would mean to apply individual-based models at any scale, which is not feasible for computational limitations. Thus, we rely on scaling if we want to model forest dynamics over larger areas.

Based on the results of this study, we advise to consider the spatial scale in model-data comparisons, when dealing with aggregated quantities such as biomass and carbon fluxes. If possible, frequency distributions should be derived and compared at several scales. Scaling relationships like the ones identified in this study may help to transfer distribution parameters between scales.

Models often have inherent scales, i.e. they represent certain processes on a grid. This is not exclusive to simple forest models like the one used in this study, but also individual-based forest models, which also simulate many processes and environmental drivers like mortality, disturbances, climate and soil using grid cells. We have shown that for reproducing the correct output

distributions (biomass) at different scales, the input distributions (gain, mortality), i.e. the model parameters, had to be rescaled accordingly.

## 4.5 Outlook

For a deeper understanding of the scaling coefficients of the different processes at the small scales below 1 ha, further analyses should look into the spatial patterns and mechanisms at the individual level, i.e., tree positions and their size and biomass distribution. Methods such as point process models (Lister and Leites, 2018), spatially explicit individual-based forest models (Maréchaux and Chave, 2017) or network analysis (Schmid et al., 2020) can improve our understanding of how patterns at the individual level shape the scaling at aggregated levels. Beyond the 1-ha scale, where environmental gradients determine the distributions, their influence on scaling can be investigated with a combination of forest landscape models (He, 2008), high resolution and multitemporal remote sensing (Dalponte et al., 2019) and machine learning based surrogate models (Rammer and Seidl, 2019).

Several applications become possible if the scaling of biomass distributions and their relation to growth and mortality are well understood. For example, Williams et al. (2013) used this to estimate disturbance intensities from remotely sensed biomass distributions using a forest model. A detailed understanding how canopy height changes at a certain scale relate to biomass changes, can allow for a direct quantification of the net forest carbon changes from remote sensing (Hiltner et al., 2022; Knapp et al., 2018b). Knowledge about the scaling of forest attribute distributions is also required for downscaling of gridded maps (i.e., super-resolution), for purposes such as data assimilation (Hill et al., 2011; Rödig et al., 2017) or pixel-to-point comparisons between models and field data (Rammig et al., 2018).

## 5 Conclusions

The study has shown how the distributions of variables, which are important for the carbon budgets of forests, vary with spatial scale. Biomass and its gain, loss and mortality could all be described with parametric distribution functions (gamma or lognormal) of varying spread at different spatial resolutions. The spread of these distributions, in the form of standard deviations, was described as a function of scale using power law relationships. Scaling exponents for the extensive properties biomass and gain were close to the expected value of -0.5, but not precisely, which indicates subtle spatial patterns in the data, while the scaling exponent of the intensive property mortality was quite different. The scaling relationships allowed for successful up- and downscaling of the respective distribution functions in the range of scales between 10 and 100 m. Beyond this range, a comparison with lidar data showed deviations from the scaling relationship. Thus, we conclude that the distributions in the considered range are dominated by the process heterogeneity (forest dynamics), while above, landscape heterogeneity plays an increasing role. Forest models need to account for this landscape heterogeneity when being applied at coarser scales. The application of up- and downscaling for forest models was demonstrated. It was shown how the scaling

relationships can be used to reproduce biomass frequency distributions with two different simple forest models across scales using measured parameters about gain and mortality from a single reference scale as input. The two models differed with regard to which scale was the best reference scale. Optimal scales always depend on the considered processes and need to be identified in every model. Scaling approaches will hopefully facilitate the comparison and trans-scale integration of data about forest dynamics from various sources of information, such as forest inventory, remote sensing and modeling.

### Code and data availability

A publicly available forest inventory dataset was analysed in this study. It can be found on Dryad: https://datadryad.org/stash/dataset/doi:10.15146/5xcp-0d46. The BCI lidar dataset is publicly available from the Office of Bioinformatics, Smithsonian Tropical Research Institute. The R code for the analyses and simulations is available upon request from the corresponding author.

### Supplement

### Author contributions

Conceptualization, N.K., S.A. and A.H.; methodology, N.K., S.A. and A.H.; software N.K.; formal analysis, N.K.; visualization, N.K.; funding acquisition, S.A. and A.H.; supervision, A.H.; writing – original draft preparation, N.K.; writing – review and editing, N.K., S.A. and A.H.; All authors have read and agreed to the published version of the manuscript.

### Competing interests

The authors declare that they have no conflict of interest.

### Acknowledgments

We would like to thank the Smithsonian Tropical Research Institute for providing the census data for BCI. The BCI forest dynamics research project was made possible by National Science Foundation grants to Stephen P. Hubbell: DEB-0640386, DEB-0425651, DEB-0346488, DEB-0129874, DEB-00753102, DEB-9909347, DEB-9615226, DEB-9615226, DEB-9405933, DEB-9221033, DEB-9100058, DEB-8906869, DEB-8605042, DEB-8206992, DEB-7922197, support from the Center for Tropical Forest Science, the Smithsonian Tropical Research Institute, the John D. and Catherine T. MacArthur Foundation, the Mellon Foundation, the Small World Institute Fund, and numerous private individuals, and through the hard work of over 100 people from 10 countries over the past two decades. The plot project is part of the Forest Global Earth Observatory (ForestGEO), a global network of large-scale demographic tree plots. We thank J. Dalling for providing the lidar

data from BCI. This study was conducted with funding from the Helmholtz Association of German Research Centres (HGF, grant number ZT-I-0010) in the Reduced Complexity Models project (RedMod).

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
