# Peer review of "A question of scale: modelling biomass, gain and mortality distributions of a tropical forest"

_Biogeosciences, 2022_

## Author Response (AR2)

**Response letter**

Thank you for the valuable feedback and the opportunity to revise our manuscript. We have now prepared a revised version. The main changes concern:

1) Revision of the Abstract and Introduction to specify the problem
2) Revision of the Discussion and Conclusion to put the findings into perspective
3) Clarifications about various aspects raised by the reviewers and choices made by the authors.

In the following we present the point by point responses for every comment by the reviewers and the editor with line numbers referring to the track changes version of the revised manuscript.

**Reviewer 1**

This manuscript considers the issue of property scaling with respect to spatial scale. In particular, the authors examine how the aboveground biomass distributions in forests change depending on the scale they are measured at. Such information may have important implications for global models that incorporate site-level data and run with pixels covering hundreds of square kilometers.

Thank you for reviewing our manuscript and for the valuable feedback. In the following, we will address all your points. Lines in our responses refer to the track changes version of the manuscript.

My thoughts on the manuscript can be summarized in three categories, and I have the feeling they are strongly related to the intendend audience. A journal like Biogeosciences has a bit wider audience than some other journals, which means that folks will be approaching it from different backgrounds. Indeed, this article is meant to address folks from several different backgrounds. As someone with a modeling background, I may have missed what is evident in other fields. To whit:

1) I did not find the problem well-demonstrated

Thank you for this important point. We have revised the Abstract and Introduction, added a section in the Introduction to demonstrate the problem better (L. 94-103) and also reformulated the research questions (L. 114-126). More details follow in the response to the long comment about point (1) further below.

2) I felt that the discussion was not well-developed enough to convey the significance of the results and convince a general reader

Thank you. We have revised the Discussion and added Perspectives and Outlook sections to put the results into context (L. 448-505). We also added explanations, which we describe the response to the long comment about point (2) further below. We also revised the Conclusions (L. 510-526).

3) I was not convinced that the method was successful compared to the case of not applying a scaling factor

Thank you. We provided additional information to address this point, which we explain in detail in the response to the following long comment about point (3).

For the last point (3), this seems easy to address by showing a figure like Figure 7 but replacing the green historgrams and the blue line with the results of the unscaled distribution. Would this just be the existing green histograms at 50m? If so, I would really appreciate somehow making this more clear (ideally in the figure, but also adding text would be good). I feel like Figure 7 is the figure showing the method was successful, but I do not see that immediately.

Thank you for this comment. The unscaled simulated distribution is indeed the green histogram at 50-m scale. We have added "Unscaled forest model" in the graphic and text to make this clear. (L. 330-331)
With Fig. 7 we tried to show the best functioning case in the main manuscript, which was the one starting with fitted parameters at 50-m reference scale. It was impossible to include graphics for all different simulation cases. Thus, we provide their results in tables S1 and S2. To show some detailed graphics about each simulation and result aggregation scale, we provided S5 to S8 (which we now mention more prominently in the text, L. 341-342), which are still only the best cases of each reference scale. This already shows, that even for the best cases and using the scaling relationship, the simulated distributions can diverge remarkably from the field distributions.

We did not present a simulation case without applying any scaling coefficient (i.e., applying scaling coefficient -0.5, according to your point 3), since rescaling with -0.5 already led to drifts in the input distributions (AGB gain and mortality), when we tested it against the field data. We added a new graphic in the supplements to clarify this point (new S3).

Figure 7 amazed me. I was shocked to see how the distributions shifted. However, I don't think I should be on page 15 of an article before I'm intrigued by it. I feel like the right hand side shows why this issue is important, and relates to my point (1) above. The introduction of the problem seems to occur in lines 75--7 with a single citation (Wong, 2008). In my mind, this should be an entire paragraph to emphasize the point: "Models which fit biomass distributions at 10 m^2 spatial resolution and reproduce them perfectly at 100 km^2 are incorrect." However, this represents a Catch-22. Phrasing the problem this way makes it much more appropriate for Biogeosciences, but would also require more evidence in the case of land surface models. However, the authors could (and I believe, do) demonstrate this problem with two simple models. Therefore, the information seems to be already present and just needs some restructuring to be more evident and grab the eye of a non-specialist (which is the case with the vast majority of Biogeosciences readership). More citations to the last sentence of the paragraph on line 80 ("But it is often unclear how scale affects observed and simulated distributions"), in particular with regards to forest plot and larger area modeling related to the carbon cycle, would be very welcome for point (1).

Thank you. We expanded the respective paragraph in the Introduction, to state the problem more clearly (L. 91-103). We also rephrased the research questions and mention the context of model-data comparisons (L. 114-126). We agree that the model results in Fig. 7 appear late in the manuscript. However, the field derived distributions and the scaling relations derived from them already demonstrate the problem earlier. The field derived distributions were essential part of the analysis and necessary input to the model. Hence, we don't see how the models can appear earlier in the story. In the new Perspectives section, we further discuss the issue (455-471).
We have added citations to the mentioned sentence (L. 111-113).

For point (2), it was not clear to me why the standard deviations are different. Figures 5-7 show that they are, but I don't understand why this happens. Section 4.2 mentions that different fitting approches had different levels of success, and explains what these fits where, but it does not explore why they had different levels of success. Is there something about the underlying data or problem which means this could have been foreseen from the beginning?

We were not sure whether this comment refers also to the differences between SDs at different scales per se, or to the differences between fitting methods. We have added text to explain the differences between scales (L. 94-103, L. 364-365, L. 455-466) and about the differences between fitting methods (L. 397-403). Biomass gain was the variable which was the most difficult to fit with the parametric distribution functions, as its distribution apparently conforms the least to the tested lognormal and gamma distribution shapes. However, we do not see how this could have been foreseen.

Minor comments:

Line 80 and 81: Perhaps "extends" should be "extents"

Thank you. We corrected it (L. 109).

Line 170: It seems that mortality modeling presents an issue with respect the scale. The simulation model chosen resets the area of a whole grid cell to zero. For a 10m x 10m

pixel, this could be a single very large tree. For a 100m by 100m pixel, this seems like a larger event. Biomass gain, on the other hand, seems to be similar for every size of pixel (if a 100 m^2 plot grows by 100 g C m-2 yr-1, then either one trees grows like that on a small pixel or it's spread among many trees on a larger pixel). Does this difference in behavior have something to do with why the simulation results change depending on pixel size?

In the original model by Fisher et al. (2008) a mortality event was indeed modelled by setting the biomass of the pixel to zero. However, such a (stand replacing) approach is not applicable across multiple scales. Therefore, we changed the model by drawing mortality from a continuous distribution. We have added text to make this clearer (L. 209-212). Thus, this is not the reason for the differences between scales. Also for biomass gain we observe the scale dependence of the frequency distribution (see Fig. 4 center column), due to growth differences between trees and their spatial positions.

Table 2: The number of signficiant figures used seems almost excessive. Is there truly rationale for mean OVL of 0.883 and 0.887? I guess if the error bars on the distribution are taken into account, the mean OVL will fluctuate by much more than that. Although the bins are big enough that the measurement errors are likely small. I would be happy if the authors could confirm this for me (a non-experimentalist).

Thank you. We think it is common to provide percentages with one additional decimal digit, which is equivalent to three digits if given as a fraction. From a practical stand point the third digit allowed us to better select the best fitting method for each case (which was most relevant in Table S2 with several very similar values). We agree that small differences in OVL are not necessarily meaningful or significant and might change if data slightly changes. For this reason, we show the whole table with all the different values to present also the "almost best" cases.

Figure 7: Please add, "On the left are the G and M distributions used as input" or something similar to clarify what the left side of the plot is for the reader.

Thank you. We added it to the caption. (L. 336)

Line 326: The line begining with "Theory states that the SD" implies to me that there is rationale behind this. I would appreciate this rationale being expressed more in the introduction to introduce the reader to the fact that this is a well-known problem with both observational and theoretical background. Perhaps it is mentioned in the Wong, 2008 reference, but adding a couple sentences would be welcome. The same for the fact that the mean is stable across all scales (line 338), which indicates it really is just an issue with the standard deviation.

Thank you. We added information about it in the introduction (L. 98-103) and the discussion (L. 364-365, 455-462).

**Reviewer 2**

The paper "A question of scale: modeling biomass, gain and mortality distributions of a tropical forest" is an attempt to explore the relationship of forest dynamic main characteristics i.e. biomass stock, biomass growth and mortality, across spatial scales between 10m to 100m. The authors used different approaches based on multi-scale observation sources and they estimated scale factors to upscale or downscale the distribution of the forest dynamic characteristics. In addition, the authors make use of stochastic simulation forest models in order to retrieve the observed distributions of each scale based only on one of them with success.

This study is overall well crafted and the material and method is particularly well written with clear statements that will help readers to reuse their works in different forest ecosystems across the globe. Nonetheless, the limited range of scales that they really used in the study (10m – 100m instead of the full range 10m-500m) reduced the impact of the study.

Thank you for reviewing our manuscript and for the valuable feedback. In the following, we will address all your points. Lines in our responses refer to the track changes version of the manuscript.

I have few general comments :

The introduction is somewhat difficult to follow because it looks like an enumeration of facts without any logical link helping the reader to follow the thinking of the authors. I would recommend using more linking words to structure the introduction and especially the first paragraph.

Thank you for this comment. We have revised the introduction and included the recommended linkages between the paragraphs (L. 51-103).

The overall method is clear but why the authors didn't use higher scaling factors such as 200m, 500m and 1000m ? The lidar survey gives the authors a way to validate them, isn't it ? If I understand well, one can extrapolate (even if the lidar approach shows divergence) upscale distributions from the log/log scaling relationship for G and M. If not, the authors must justify their choice in section 2.3.

Thank you. We have chosen to focus on the scales between 10 and 100 m as we consider the frequency distributions between these scales being primarily demography driven, while at larger scales they are driven by environmental gradients at landscape scale. The lidar analysis shows the deviation of the scaling relationship, but it also shows how quickly even the whole island becomes too small to obtain enough data records for analysis. These landscape gradients were however not represented in the model. When we applied the model at scales coarser than 100 m we obtained increasingly narrow, and increasingly normally distributed biomass distributions which further follow the scaling relationship. But since this is not what we observe in real landscapes, we did not consider it valuable to show model results beyond the 100-m scale. We added the explanation in section 2.3 (L. 175-178).

In the result section, again, I found the figure 4 a bit disturbing since most of the study relates on a range of scale between 10m to 100m e.g. scaling factors are calculated for 10m, 20m, 50m and 100m. Modeling section is also made between 10m to 100m. I would recommend choosing between including the larger range in both modeling and scaling factor sections (which may lead to less clear results but will increase the paper's impact) or put the lidar analysis in supplementary material in order to clearer the message (but decrease the paper's impact).

Thank you for these suggestions. We have decided for the second option to move Fig. 4 to the supplements (now Fig. S4), as we have explained above, that the analysis of the effect of landscape gradients on scaling was beyond this study and not included in the model. The lidar analysis was meant as a first step towards looking beyond the 100-m scale, but we think modelling landscape heterogeneity would be a topic for another study.

We have added statements about the influence of landscape heterogeneity in the discussion (L. 387-390, 424-427, 513-517).

The discussion about the technical aspect is good but, at line 365, the sentence about the issue on the weak performance of simulations using 100-m reference gives no information at all on what would be the cause of this issue. Discussion is exactly the place where the authors can give their thoughts about it. So please, share with the readers otherwise it feels like the authors want to hide something.

Thank you. We added text about what we think the cause might be (L. 418-427).

We wait for this section to lighten us on how the author's work will benefit others (modelers colleges but also no-modelers). I found the section a bit vague without practical examples. I also would like to read a perspective section in which the reader will know more about what the authors are planning in order to solve issues regarding the weaknesses they found during their study.

Thank you. We have added a perspectives (L. 448-489) and an outlook section (L. 490-505).

**Editor**

**Comments to the author:**

Thank you for submitting a revised version of your manuscript. In the light of the referee comments I found the revisions insufficient to send the revised manuscript to the referees. Based on the original submission both referees expressed their concern about the presentation of the work and struggled to understand the implications of this study. After reading the entire manuscript, I still share the initial concern of the referees. The study does a good job in explaining to methods and the results. However, following revisions, the abstract (no revisions), introduction, and perspective sections do not help the reader to understand the significance of this study. A thoroughly revised manuscript will be send out to the same referees.

Thank you for the constructive feedback and for giving us the opportunity to revise the manuscript. We have revised the Abstract, Introduction, Discussion and Conclusion according to the comments. Lines in our responses refer to the track changes version of the manuscript.

**Major concerns**

The abstract is the first item prospective readers will check. The current abstract does a poor job in explaining the problem as well as giving a perspective on how the results can be used by others (see also below). Rewrite the abstract with a focus on the problem and solutions such that it convinces prospective readers to read the whole paper. In this regard, the last sentence of the abstract would make for a good first sentence of the abstract.

Thank you. We have rewritten the abstract following the suggestions (L. 13-22, 34-38).

Nowadays, too many forest-studies start with a sentence similar to your first sentence (i.e. "The quantification of forest carbon budgets is important for understanding the role of forests in the global climate system" and "Forests are an important pool in the global carbon cycle"). Such a sentence will not help to draw the attention of the readers. Moreover, the authors refer several times to the C-cycle but these references are superficial and do not really connect to the results (biomass is only one aspect of the C-cycle). As scaling issues extent beyond the C-cycle consider broadening the introduction.

Thank you. We have removed the C-cycle from the Abstract and Introduction and replaced it with more broad formulations (L. 13-22, 41-50).

In the introduction three objectives are listed. Ideally those objectives should be used to structure the discussion. However, the objectives in this manuscript are too technical to use them as a compelling structure for the discussion. Try to rephrase these questions such that they have a clear link to the overall challenge "compare data and simulations with a very different resolution". At present it is not clear for the readers why objective 3 is important. If you manage to rephrase these questions and restructure the discussion there might no longer be a need for a perspective section.

Thank you. We are now mentioning the overall challenge of model-data comparisons and have rephrased the questions, such that they are being answered in the discussion (L. 114-126).

The methods sections as well as the result section are good. In comparison to the results, the discussion should provide additional insights and interpretation. The current discussion fails to place the results in a wider context based on previous literature on the topic. The discussion should show which results confirm previous insights and which results provide new insights. Subsequently, the implications of these new insights should be described.

Thank you. We have revised the discussion (L. 263-369, 441-505).

A substantial part of the discussion and perspective describes what can be done in the future. If you find this important, please, move all of this information in a section called "outlook" or "future work". The discussion should focus on the results that you obtained not on the results that you did not obtain. Rather than summarizing the discussion, the perspective should help the readers to understand the implications of this study. From this regard, the first paragraph of the current perspective section is useless.

Thank you. We have revised the perspectives section and moved the aspects about possible future work to the new outlook section (L. 448-505).

At present the manuscript well presents the technical aspects of a detailed local study. It nicely demonstrates the possible challenges for data-model comparison, and tests different methods but it remains unclear what you learned from this study. How are you going to use the results from this study to improve your model? How are you going to use these results to improve data-model comparisons? What can other model groups do with these results? Does every model needs to determine its inherent resolution? What is the take home message of this study? How does your results relate to using (very) high-resolution remote sensing products?

Thank you. We do not have answers to all these questions, but we hope that our revisions could clarify some of them.

**Minor comments**

L36 "For this reason, the worldwide quantification of forest biomass stocks and changes are important." The this now refers to production and mortality and the sentence is no longer logic. Rewrite.

Thank you. We have removed this sentence (L. 46).

L71 "and the past" replace by "in the past"

Thank you. We have removed this part (L. 87).

L367 Replace "which led" by "which led to "

Thank you. Corrected (L. 391).

L400 this sentence needs a reference to support the 100 m threshold. Which edaphic factor would drive this 100m increase in landscape heterogeneity.

Thank you. We do not think of 100-m of a threshold, but rather as a scale where the individual trees play less of role, hence the influence of landscape heterogeneity increases relatively. We have rephrased the sentence accordingly and added a reference about the landscape heterogeneity in BCI (L. 387-390, 424-427).

L433 This statements needs a quantitative reference. Delete this sentence if it cannot be backed up by a review study.

Thank you. We have rephrased and added a reference (L. 449-450).

L434 Start the perspective with this paragraph. This gives a clear idea of why this study is needed. Paragraphs explaining what other modelers should learn from this study is still missing. Is there something experimentalists and the remote sensing community can learn from this study? Or is the whole burden on the modelers?

Thank you. We have moved the paragraph to the beginning of the perspective section (L. 449-454) and have added advices for modelers (L. 439-447, 462-471, 512-521).

L439 Better explain your thinking. Difficult to see a link with the results of this study.

Thank you. We have added a sentence to clarify the link (L. 499-500).

L445 "Hence, with the advancement of forest monitoring and modeling the demand for consistent scaling methods is increasing." Yes I agree. This manuscript studies scaling but it is unclear whether it succeed in providing consistent scaling methods. If so, this should be explicitly stated and linked to the results.

Thank you. We hope that with our revisions the link between the results and conclusions has become clearer.

**Notification to the authors**:

Please rename the figure #54 to figure #5. 2. Please ensure that the colour schemes used in your maps and charts allow readers with colour vision deficiencies to correctly interpret your findings. Please check your figures using the Coblis – Color Blindness Simulator (https://www.color-blindness.com/coblis-color-blindness-simulator/) and revise the colour schemes accordingly.

Thank you. Figure 5 is now correctly numbered. We have recreated several figures using colourblind-friendly palettes from the R package viridis (S1, S2, S3, S4) and using colours of very different brightness for histogram overlays (6, S5, S6, S7, S8) and points (S4e). We checked all with Coblis. The majority of figures in the main manuscript was kept black and white for colourblind friendliness. In Fig. S9 we added "upper row" and "lower row" to the caption and the colours do not provide any additional information.